# Transcriptome Profiles Reveals *ScDREB10* from *Syntrichia caninervis* Regulated Phenylpropanoid Biosynthesis and Starch/Sucrose Metabolism to Enhance Plant Stress Tolerance

**DOI:** 10.3390/plants13020205

**Published:** 2024-01-11

**Authors:** Yuqing Liang, Xiaoshuang Li, Feiya Lei, Ruirui Yang, Wenwan Bai, Qilin Yang, Daoyuan Zhang

**Affiliations:** 1State Key Laboratory of Desert and Oasis Ecology, Key Laboratory of Ecological Safety and Sustainable Development in Arid Lands, Xinjiang Institute of Ecology and Geography, Chinese Academy of Sciences, Urumqi 830011, China; lyuqing007@ms.xjb.ac.cn (Y.L.);; 2Turpan Eremophytes Botanical Garden, Chinese Academy of Sciences, Turpan 838008, China; 3Conservation and Utilization of Plant Gene Resources, Key Laboratory of Xinjiang, Urumqi 830011, China; 4University of Chinese Academy of Sciences, Beijing 100049, China

**Keywords:** AP2/ERF transcriptional factor, *ScDREB10*, phenylpropanoid biosynthesis, starch and sucrose metabolism, *Syntrichia caninervis*, plant stress tolerance

## Abstract

Desiccation is a kind of extreme form of drought stress and desiccation tolerance (DT) is an ancient trait of plants that allows them to survive tissue water potentials reaching −100 MPa or lower. *ScDREB10* is a DREB A-5 transcription factor gene from a DT moss named *Syntrichia caninervis*, which has strong comprehensive tolerance to osmotic and salt stresses. This study delves further into the molecular mechanism of ScDREB10 stress tolerance based on the transcriptome data of the overexpression of *ScDREB10* in *Arabidopsis* under control, osmotic and salt treatments. The transcriptional analysis of weight gene co-expression network analysis (WGCNA) showed that “phenylpropanoid biosynthesis” and “starch and sucrose metabolism” were key pathways in the network of cyan and yellow modules. Meanwhile, Gene Ontology (GO) and Kyoto Encyclopedia of Genes and Genomes (KEGG) analysis of differentially expressed genes (DEGs) also showed that “phenylpropanoid biosynthesis” and “starch and sucrose metabolism” pathways demonstrate the highest enrichment in response to osmotic and salt stress, respectively. Quantitative real-time PCR (qRT-PCR) results confirmed that most genes related to phenylpropanoid biosynthesis” and “starch and sucrose metabolism” pathways in overexpressing *ScDREB10 Arabidopsis* were up-regulated in response to osmotic and salt stresses, respectively. In line with the results, the corresponding lignin, sucrose, and trehalose contents and sucrose phosphate synthase activities were also increased in overexpressing ScDREB10 *Arabidopsis* under osmotic and salt stress treatments. Additionally, cis-acting promoter element analyses and yeast one-hybrid experiments showed that ScDREB10 was not only able to bind with classical cis-elements, such as DRE and TATCCC (MYBST1), but also bind with unknown element CGTCCA. All of these findings suggest that ScDREB10 may regulate plant stress tolerance by effecting phenylpropanoid biosynthesis, and starch and sucrose metabolism pathways. This research provides insights into the molecular mechanisms underpinning ScDREB10-mediated stress tolerance and contributes to deeply understanding the A-5 DREB regulatory mechanism.

## 1. Introduction

Extreme environmental conditions such as drought, high salinity, and extreme temperatures exert profound impacts on the natural distribution of plants, impede their growth and development, and ultimately lead to low agricultural yields [1,2]. Plants respond to these environmental stresses by undergoing a series of physiological changes, including alterations in cellular membrane composition, inhibition of photosynthesis, generation of reactive oxygen species (ROS), and closure of stomata [3]. Notably, a common response to such stressors involves the upregulation of specific stress-related genes, particularly transcription factors (TFs). These TFs serve as critical mediators in the intricate regulatory networks governing the transcription of downstream stress-responsive genes [4,5]. In the realm of plant biology, there exist approximately 58 distinct TF gene families (as documented in the Plant TFDB5.0 database [6]), with certain gene families playing pivotal roles in modulating both biotic and abiotic stress responses. Among these, the AP2/ERF TF superfamily emerges as one of the largest and most influential families influencing stress tolerance in plants [7].

The AP2/ERF (APETALA2/Ethylene-Responsive Factor) gene family stands out as one of the most extensive gene families exclusively found in plants. Among the members of this family, DREB (dehydration-responsive element binding) proteins, a subfamily of *AP2/ERF* genes, have been identified as crucial regulators of abiotic stress tolerance [8,9]. Recently, research has revealed numerous DREBs in non-model plants, highlighting their diverse roles in stress tolerance, signaling pathways, growth, development, and secondary metabolism [10,11,12,13]. The DREB subfamily can be further categorized into six distinct subgroups, denoted as A-1 through A-6. While the functions of the A-1 and A-2 subgroups have been extensively explored [14], the roles and underlying mechanisms of A-5-type DREB genes have garnered limited attention and remain relatively enigmatic, despite their promising potential. In recent years, of the majority of studies have reported that A-5 DREB transcription factors can significantly enhance stress tolerance in plants [15,16,17,18,19,20,21]. Collectively, these investigations highlighted the potential of A-5 *DREB* genes as promising candidates for crop breeding programs aimed at enhancing stress tolerance. Thus, it is necessary to investigate the biological functions of A-5 *DREB* genes across diverse plant species.

*Syntrichia caninervis* serves as an excellent model for comprehending the physiological, biochemical, and molecular factors associated with desiccation tolerance, as well as a potential source of target genes for crop improvement [22,23,24,25]. A total of ten *DREB* genes (*ScDREB1*–*ScDREB10*) were screened and cloned from the desert moss *S. caninervis* based on the transcriptome of *S. caninervis* during dehydration–rehydration [26]. We functionally validated these ten genes and found that overexpression of *ScDREB10* led to significant enhancements in drought and salt tolerance in transgenic *Arabidopsis* plants, both during germination and at the seedling stage [27]. Notably, ScDREB10 exhibited comprehensive mechanisms for tolerance, including enhanced scavenging of reactive oxygen species (ROS) and the upregulation of stress-related genes, contributing to improved osmotic and salt stress tolerance in *Arabidopsis* [27]. To gain deeper insights into the molecular and functional mechanisms and modulate the physiological pathways regulated by ScDREB10, this current study utilized transcriptome data from *Arabidopsis* plants overexpressing *ScDREB10*. We conducted comprehensive analyses, including the identification of genes differentially expressed (DEGs), Weighted Gene Co-expression Network Analysis (WGCNA), Gene Ontology (GO) and Kyoto Encyclopedia of Genes and Genomes (KEGG) pathway enrichment analyses, and the exploration of enrichment of cis-acting promoter elements. Two pathways of particular significance were further investigated. Our findings have revealed the potential of ScDREB10 in orchestrating crucial physiological pathways aimed at enhancing osmotic and salt tolerances. Importantly, we have identified unique cis-elements that bind with ScDREB10, shedding light on its regulation of different downstream stress-responsive genes.

## 2. Results

### 2.1. The DEGs Analysis of Transcriptomes between Overexpression-ScDREB10 and WT Plants in Response to Osmotic and Salt Stresses

RNA sequencing (RNA-Seq) analysis was conducted to explore the potential roles of ScDREB10 in transcription regulation and identify pathways influenced by ScDREB10. The control, osmotic stress, and salt stress samples from both wild-type (WT) and overexpression-*ScDREB10* line 6 plants were collected for RNA-Seq analysis (Figure 1a). Under osmotic stress, WT plants exhibited 1323 up-regulated DEGs and 1112 down-regulated DEGs (WT osmotic stress vs. WT control, Figure 1b). In contrast, the overexpression-*ScDREB10* lines displayed a higher number of DEGs, with 2089 up-regulated DEGs and 789 down-regulated DEGs (OE osmotic stress vs. OE control, Figure 1b). Similarly, under salt stress, overexpression-*ScDREB10* lines showed 1711 up-regulated and 1294 down-regulated DEGs (OE salt stress vs. OE control, Figure 1b). Comparing the transgenic line 6 with WT, 1053 up-regulated and 395 down-regulated DEGs were detected under control conditions (Figure 1d). However, under osmotic and salt stresses, transgenic line 6 exhibited 2137 up-regulated and 968 down-regulated DEGs, as well as 1453 up-regulated and 1086 down-regulated DEGs, respectively (Figure 1d). Overlapping analysis revealed that 654 up-regulated and 486 down-regulated DEGs were commonly found in both osmotic and salt stress in WT (Figure 1c). Similarly, transgenic line 6 exhibited 983 up-regulated and 372 down-regulated DEGs in common between osmotic and salt stresses (Figure 1c). Similar trends are illustrated in all three comparisons (Figure 1e). A comprehensive list of all DEGs can be found in Appendix A. We observed that the overexpression of ScDREB10 resulted in a greater modulation of gene expression under osmotic stress compared to salt stress and normal conditions, particularly in terms of up-regulated genes and a lower number of down-regulated genes, as compared to the WT under osmotic stress.

### 2.2. GO and KEGG Analyses between Overexpression-ScDREB10 and WT Transcriptomes in Response to Osmotic and Salt Stresses

In order to investigate the roles of ScDREB10 in osmotic or salt stress tolerance, the functional annotation of the up-regulated DEGs under control, osmotic and salt treatments by GO classification enrichment analysis (Figure 2a). In total, 30 terms were chosen according to the top 16 terms with the padj < 0.05 under control, osmotic and salt stress conditions (*ScDREB10*-OE vs. WT). Six GO terms that referred to “tetrapyrrole binding”, “peroxidase”, “heme binding”, “external encapsulating structure organization”, “cell wall organization or biogenesis”, and “ATPase activity” were significantly enriched in all treatments, and more enriched in osmotic and salt stresses when compared with the control treatment (Figure 2a). GO terms “tetrapyrrole binding” and “heme binding” were the most significantly enriched in three treatments. Top ten KEGG pathway enrichment analysis of up-regulated DEGs under control, osmotic and salt treatments showed that “phenylpropanoid biosynthesis” was the most enriched in osmotic stress, while “starch and sucrose metabolism” was the most enriched in salt stress when compared with the control (Figure 2b). In conclusion, GO and KEGG analysis revealed that these up-regulated DEGs were involved in various biological processes.

### 2.3. WGCNA of All DEGs between Overexpression-ScDREB10 and WT Transcriptomes in Response to Osmotic and Salt Stresses

Based on the transcriptome data of overexpression-*ScDREB10* and WT transcriptomes under control, osmotic and salt stress treatments, a total of 9919 DEGs among different samples were selected for WGCNA. Co-expression network analysis identified 17 modules, in which 2 modules were significantly associated with ScDREB10 in response to osmotic and salt stresses (cyan positively correlated with OE vs. WT in response to osmotic stress; yellow positively correlated with OE vs. WT in response to salt stress) (Figure 3a,b). The cyan and yellow modules consist of 665 and 389 DEGs, respectively; to identify the key pathways of these DEGs, we performed KEGG enrichment analysis. The “phenylpropanoid biosynthesis” and “starch and sucrose metabolism” were significantly enriched (*p* < 0.05) in cyan and yellow modules (Figure 3c,e). The KEGG pathway enrichment network for cyan and yellow modules also indicated that the “phenylpropanoid biosynthesis” and “starch and sucrose metabolism” were key pathways in the network (Figure 3d,f). The “phenylpropanoid biosynthesis” pathways were 26 and 11 DEGs enriched in cyan and yellow modules respectively, and “starch and sucrose metabolism” pathways were 15 and 11 DEGs enriched in cyan and yellow modules, respectively (Figure 3d,f).

### 2.4. ScDREB10 May Participate in Regulating Phenylpropanoid Biosynthesis in Response to Plant Osmotic Stress

The RNA-seq data from *Arabidopsis* suggested that ScDREB10 might have a role in enhancing osmotic resistance and promoting lignin biosynthesis by regulating phenylpropanoid metabolism. This hypothesis is supported by the significant up-regulation of genes involved in phenylpropanoid biosynthesis in the KEGG enrichment analysis. Among the key enzymes associated with lignin biosynthesis, including PALs, 4CLs, CADs, CCRs, and PERs, most of them in the ScDREB10 transgenic line were found to have similar or lower abundance levels compared to the WT under normal conditions. However, under osmotic stress, the expression levels of 11 genes in the overexpression-*ScDREB10* line were significantly higher than those in the WT (Figure 4a and Appendix A). Additionally, the expression levels of *F5H* and *COMTs* genes, which are involved in different lignin forms, were also altered (Figure 4a and Appendix A). To validate the findings from the RNA-Seq analysis, quantitative real-time PCR (qRT-PCR) was performed on eight genes related to lignin synthesis in the overexpression-*ScDREB10* lines and WT under normal conditions and osmotic stress (Figure 4b). No significant differences were observed between the overexpression-*ScDREB10* lines and WT under normal conditions. However, under osmotic stress, the expression levels of most genes in the transgenic plants were significantly higher compared to the WT. Notably, *PAL1, 4CL2*, and *CAD9* in the overexpression-*ScDREB10* lines showed a substantial increase in expression, approximately 2.5-fold, 4-fold, and 5.5-fold, respectively, compared to the WT (Figure 4b). The consistency between the RNA-Seq and qRT-PCR data for these genes further confirms the reliability of the RNA-Seq analysis. Furthermore, the lignin content was measured in both the WT and transgenic lines under normal conditions and osmotic stress. The results revealed no difference in lignin content between the WT and transgenic plants under normal conditions. However, under osmotic stress, the transgenic plants exhibited significantly higher lignin contents compared to the WT (Figure 4c).

### 2.5. ScDREB10 May Participate in Regulating Starch and Sucrose Metabolism in Response to Salt Stress

According to the analysis of KEGG pathway enrichments, the category that showed the highest enrichment in response to salt stress was “starch and sucrose metabolism”. Significant upregulation of genes related to starch and sucrose metabolism was observed in the transgenic line after exposure to salt stress (Figure 5a). In order to gain a deeper understanding of how the overexpression of *ScDREB10* enhances salt tolerance in *Arabidopsis*, we examined the changes in genes responsible for encoding the enzymes involved in starch and sucrose biosynthesis. Comparing the overexpression-*ScDREB10* lines with the WT under normal conditions, an increase in the expression of genes encoding enzymes such as *INVs*, *SPPs*, *SSs*, *SPSs*, and *APLs* was observed in the transgenic lines. A total of 23 genes were enriched in the starch and sucrose metabolism pathway, and their expression level significantly increased in the overexpression-*ScDREB10* line compared to the WT under salt stress (Figure 5a and Appendix A). Two genes were up-regulated in the trehalose synthesis process after salt treatment compared with the WT. The starch synthase genes (*SS3*, *SS4*, *GBSS1*) and sucrose-phosphate synthase genes (*SPS1, 2, 3, 4*) in the starch synthesis process were up-regulated in the OE line 6 under salt stress (Figure 5a). To confirm the results of RNA-Seq analysis, qRT-PCR was executed for nine genes associated with the starch and sucrose metabolism (Figure 5b). The majority of these genes exhibited comparable alterations in expression between the RNA-Seq and RT-qPCR data. Notably, following salt treatment, the expression level of these genes in OE line 6 significantly surpassed that in the WT, particularly for *CWINV1*, *TPS5*, *SPS4*, and *SS4* (approximately 4–20 times higher). To understand the effects of ScDREB10 on starch and sucrose synthesis, we analyzed the sucrose content, sucrose phosphate synthase (SPS) activity and trehalose contents in both WT and transgenic plants. The results showed that the sucrose, sucrose phosphate synthase (SPS) and trehalose contents of transgenic plants were significantly higher than that in WT for both control and salt treatments (Figure 5b–d). The results suggested that ScDREB10 could regulate the salt stress tolerance by mediating starch and sucrose metabolism pathways.

### 2.6. Cis-Acting Promoter Element Analysis of ScDREB10 in Response to Osmotic and Salt Stresses

A cis-acting promoter element analysis based on the transcriptome data and a yeast one-hybrid assay were used to assess the binding ability of the ScDREB10 to specific promoter binding site sequences. To characterize the promoters of the up-regulated genes in the overexpressing-*ScDREB10,* we assessed the enrichment of all hexamer motifs (from AAAAAA to TTTTTT, 4^6^ = 4096) in the promoter of the upregulated genes in transgenic plants (Figure 6). We compared the frequency of each hexamer sequence in the promoters of these genes with their normalized frequencies in *Arabidopsis* promoters [30]. In order to know which cis-acting promoter elements were induced by osmotic and salt stresses, we compared the significantly enriched cis-elements of three different combinations: osmotic stress versus control, salt stress versus control and osmotic stress versus salt stress (Figure 6a). Some of the TCTC-riched and TATA box cis-acting elements were enriched in the control treatment (Figure 6a and Appendix A); however, CCCCAC, GTTATT (ARR1AT), CGTCCA, TATCCC (MYBST1) and CATTT (BOXIII) ranked among the top 10 up-regulated genes under osmotic stress (Figure 6b, Appendix A). Similarly, under salt stress, for CCTACC (DRE-like) and GTGATT (ARR1AT), these cis-elements occupied the top 10 among the top 100 up-regulated genes. The CCCCAC, CCTACC (DRE-like) and GTTATT (ARR1AT) were enriched both under osmotic and salt stresses (Figure 6c, Appendix A). We selected some classical cis-elements combined by AP2/ERF and the enriched elements based on transcriptome cis-acting element analysis to carry out a yeast one-hybrid assay. The results showed that on SD/-Trp/-Leu/-His dropout medium containing 90 mmol L^−1^ 3-AT, most cells containing the reporter construct pGADT7-ScDREB10 grew well with all bait elements, and DRE, TATCCC (MYBST1), and CGTCCA (New3) grew the best, which indicated that ScDREB10 was the most likely to bind with these cis-elements (Figure 6d). 

## 3. Discussion

A number of studies have shown that the DREB subfamily mainly participates in abiotic stress tolerance [9,31,32,33,34], and increasing evidence has shown that A-5 type DREB has been reported to be mainly involved in the response to various stresses in bryophytes [18,23,26], although the situation is different in A-2 type DREB, which may be the dominant regulator in the response to drought stress tolerance for higher plants, such as in *Arabidopsis* [26]. In the desiccation-tolerant moss *S. caninervis*, most A-5 DREBs showed various levels of stress tolerance in yeast or *Arabidopsis* and ScDREB5 and ScDREB8, which belong to A-5b and A-5a type DREB, respectively, improved salt stress tolerance in *Arabidopsis* [25,27]. In a previous study, we found that overexpressing-*ScDREB10* could comprehensively enhance drought and salt tolerances by improving ROS scavenging abilities and upregulate stress-related genes in *Arabidopsis* [24]. In this study, the WGCNA, GO and KEGG pathway enrichment analysis of overexpressing-*ScDREB10* in transcriptome data showed that “phenylpropanoid biosynthesis” and “starch and sucrose metabolism” were the most enriched categories under osmotic and salt stresses, respectively, indicating the possibility that ScDREB10 might regulate the expression of genes associated with the phenylpropanoid, and starch and sucrose biosynthetic pathways. This hypothesis was further supported by the measurement of the physiological index and transcript analysis of phenylpropanoids, and starch and sucrose biosynthetic genes in transgenic *Arabidopsis* plants overexpressing ScDREB10 under osmotic and salt stresses. 

### 3.1. ScDREB10 Contributes to Drought Tolerance by Activating the Lignin Biosynthesis Pathway

Lignin biosynthesis has been studied intensively in recent decades and many studies reported that the TF-regulated lignin biosynthesis pathways were correlated with stress resistance in plants [35]. Previous studies showed that AP2/ERF family members could regulate the lignin biosynthesis genes in response to stress [36,37,38,39,40,41,42,43]. In the current study, RNA-Seq data indicated that the phenylpropanoid biosynthesis pathway was enriched in overexpression-*ScDREB10* plants under osmotic stress, which is in accordance with the high-lignin-biosynthesis-related gene expression level in OE plants under osmotic stress (Figure 4). The results are consistent with a previous study of Lee et al. (2016) [44] showing that overexpression of *OsERF71* alters drought resistance in transgenic rice by elevating the expression levels of genes related to lignin biosynthesis. In addition, transcriptome analysis revealed that OsERF83 regulates drought response genes, which are related to lignin biosynthesis genes (*OsLAC17*, *OsLAC10*, *CAD8*) [45]. Similarly, DcDREB1A in *Daucus carota* also contribute to the accumulation of lignin-biosynthesis-related gene expression and participates in the drought stress response [46]. These studies suggest that the ways in which AP2/ERF members regulate lignin biosynthesis may be common regulation methods. There are 34 genes that are involved in monolignol biosynthesis [47]. Phe ammonia-lyase (PAL) is the first enzyme to remove Phe from aromatic amino acid biosynthesis to generate trans-cinnamic acid of the phenylpropanoid pathway [48]. There are four genes that fall into two phylogenetic groups consisting of *PAL1-PAL4* in *Arabidopsis*, and the functions of these genes are different. Among this, *PAL1* and *PAL2* have been proposed to be involved in lignification [49]. In this study, the expression of *PAL1* and *PAL2* was highly induced compared to that of *PAL3* and *PAL4*, and following osmotic treatment, the expression of *PALs* in transgenic plants was higher than that in WT under osmotic stress, especially in *PAL1* and *PAL2*; the results suggested that *PALs* could be synergistically induced by a variety of environmental stimuli [21,50]. HCT is an enzyme of the monolignol pathway, catalyzing 4-Coumaroyl CoA into 4-coumaroyl shikimate. The expression of *HCT* in transgenic plants was higher than that in the WT under osmotic stress (Figure 3a and Appendix A). The genes involved in the differential pathways leading to the monolignol monomers *C3H*, *F5H*, *CCoAOMT* and *COMT* showed moderate to high expression levels under control and osmotic stress, but showed no significant differences (Figure 3a and Appendix A). CCRs and CADs were specifically involved in the biosynthetic of lignin monomers, which were involved in the conversion of esters to aldehydes, and of aldehydes to alcohols (side-chain modification), which exhibited low to moderate expression levels, but were upregulated by 1–3 fold under control and osmotic stress (OE vs. WT) (Appendix A). All these results suggested that transgenic ScDREB10 may impact the change in lignin biosynthesis process, which reflected the differences in the expression levels of the monomer biosynthesis. 

### 3.2. ScDREB10 Contributes to Salt Tolerance by Activating the Starch and Sucrose Metabolism

When plants are exposed to high salt levels, they experience osmotic stress and ion toxicity. To cope with these challenges, plants activate various adaptive mechanisms, including the accumulation of osmoprotectants such as sugars (starch and sucrose) that help maintain cellular osmotic balance and protect against ion toxicity [51]. Several studies have shown that many transcription factors can enhance salt tolerance in plants by activating the genes involved in starch and sucrose metabolism [52,53]. Sucrose-phosphate synthase (SPS) is a crucial enzyme in the cytoplasmic sucrose synthesis pathway. Its main function is to facilitate the conversion of UDP-glucose and fructose-6-phosphate into sucrose phosphate [54]. In relation to salt stress, research has demonstrated that the expression of the *SPS* gene is notably increased when there is a significant rise in sucrose levels [55,56,57,58], which suggests that *SPS* plays a key role in the regulation of sucrose synthesis under conditions of salt stress. Similarly, in our study, the expression of *ScDREB10* significantly accelerated sucrose accumulation and *SPS* was also strongly activated in the transgenic plants overexpressing *ScDREB10* (Figure 5b), which indicated that ScDREB10 was involved in the synthesis and accumulation of sucrose in *Arabidopsis*. These results showed that the overexpression of *ScDREB10* affects the sucrose biosynthesis-related genes, accelerating the accumulation of sucrose content in transgenic *Arabidopsis*. This leads to improved osmotic adjustment and ion homeostasis and maintains the energy balance, ultimately enhancing the plant’s salt tolerance. Trehalose plays a crucial role in protecting cells and organisms from various stresses and it acts as a stabilizer to help organisms survive unfavorable conditions by stabilizing proteins, membranes, and cellular structures [51,59]. Previous studies have shown that many transcription factors can enhance salt tolerance in plants by increasing the trehalose content [60,61,62,63]. Trehalose biosynthesis had been intensively investigated in plants and has been found to include two steps: the first step involves trehalose-6-phosphate (TPS), which transfers glucose from UDP-glucose to trehalose-6-phosphate and then synthesizes trehalose [64]. In the present study, the expression of *TPS* and trehalose-6-phosphatephophatas (*TPP*) in transgenic plants was significantly increased compared to that in the WT, especially under salt stress conditions (Figure 5b). In addition, under salt stress conditions, the overexpression of *ScDREB10* in *Arabidopsis* also increased the trehalose content compared with the WT (Figure 5c), indicating that ScDREB10 could induce the expression of *TPS* and *TPP* to regulate the biosynthesis of trehalose, leading to increased salt stress tolerance. These results are in line with the results of current studies about ThCRF1, which also regulated trehalose synthesis to improve salt tolerance in transgenic *Arabidopsis* [60]. In contrast, a transcription factor named OsNAC23 directly represses the transcription of *TPP1* to simultaneously elevate Tre6P and repress trehalose levels [65].

### 3.3. ScDREB10 Was Not Only Able to Bind with Classical Cis-Elements, but Also Bind with Unknown Elements

The DREB subfamily transcription factors can specifically bind to CRT/DRE elements to regulate stress-related gene expression [32,52]. In this study, we used cis-element analysis and yeast one-hybrid assay to assess the specific promoter binding site motif (total of 14) of ScDREB10 (Figure 6b). Accordingly, cis-acting promoter element analysis showed that DRE-like elements CCCCAC and CCTACC revealed a significant enrichment (*p*-value < 0.01) bound by ScDREB10 both under osmotic and salt treatments (Figure 6a, Appendix A). In addition, from the results of yeast one-hybrid assay, we found that DRE, CCCCAC and TATCCC (MYBST1) were the most likely to bind with ScDREB10 (Figure 6b). DRE is a classical cis-element that binds via DREB1A and DREB2A [66] and in our study, ScDREB10 exhibited a strong binding ability with the DRE element; although the frequency of occurrence was not very high, the results are in line with previous knowledge. In addition, cis-element and yeast one-hybrid experiment results showed that ScDREB10 could also bind with New 3 (CGTCCA), another form of AuxRR element, suggesting a new element could bind to A-5 DREB (Figure 6b). We also tested the binding ability of ScDREB10 with obo-box, which reportedly can bind to A-5 DREB ORA47 to negatively regulate ABA signaling [20]. However, in our study, the obo-box element exhibited a weak binding ability (Figure 6b). All these results suggested that A-5 DREB could bind to different elements, and thus regulate different downstream genes, which may be different from A-1 and A-2 DREBs. Moreover, our results exhibited that ScDREB10 may regulate the lignin biosynthesis, starch, and sucrose metabolism, so we addressed the role of ScDREB10 in the transcriptional regulation of lignin biosynthesis and starch, and sucrose metabolism pathways. Accordingly, the 13 motifs mentioned and most significantly enriched cis-elements mentioned above were identified in the promoters of lignin biosynthesis, starch and sucrose metabolism pathways (Appendix A). The results showed that CATTTT, AAACAA, TTTCTC, TCTTTT, and TTCTCT (New4) elements appeared at a high frequency in the promoter region of the lignin biosynthesis, starch and sucrose metabolism pathways, and appeared at least once in the promoter region of most genes (Appendix A); however, they exhibited weak bind abilities with ScDREB10 (Figure 6b). Previous studies reported that MYB TF could regulate lignin biosynthesis pathways by binding to AC elements [67,68], although studies about AP2/ERF TFs binding to elements in the promoter region of lignin biosynthesis-related genes remain limited. Ii049 from *Isatis indigotica* could regulate *IiPAL* and *IiCCR* by binding to RAA, CE1, and CBF2 [37]. ScAPD1-like from *S. caninervis* could regulate *PAL* and *C4H* by binding to RAV1, AC and CE1 [42]; in our study, combining the cis-element and yeast one-hybrid results, we found that ScDREB10 could bind to DRE, MYBST1, New1 and New4, which may be regulate genes related to lignin biosynthesis, starch and sucrose metabolism.

## 4. Materials and Methods

### 4.1. Plant Materials and Methods

*Arabidopsis* ecotype Columbia-0 (Col-0) was used as the wild-type (WT) plants. The transgenic ScDREB10 plants were previously obtained [24] and the T3 generation transgenic line 6 of ScDREB10 was selected because of its greater transcript abundance and superior performance under stress conditions compared to the WT (Li et al., 2019 [24]). The WT and overexpression-*ScDREB10* plants (line 6) were grown under normal conditions with a 16 h light/8 h dark photoperiod and 70% relative humidity at 25 °C. Seven-day-old WT and line 6 of overexpression-*ScDREB10* plants were transferred into MS medium with 300 mM mannitol for osmotic stress or 150 mM NaCl for salt stress. The seedlings transferred to MS medium were treated as the control. Tissue samples of the WT and transgenic line 6 were collected after 7 days of growth on the media and stored at −80 °C for subsequent analysis.

### 4.2. RNA Sequencing (RNA-Seq) Analysis

Control, osmotic and salt stresses of the WT and overexpression-*ScDREB10* line 6 samples were used for library construction; each sample was replicated three times. RNA sequencing analysis was performed by the Novogene Corporation (Beijing, China). After removing the adaptors and low-quality reads, the remaining reads were mapped to the reference genome of Arabidopsis thaliana (TAIR 11) using HISAT2 version 2.1.0 with the default settings [69]. The quality control and mapping rates of each sample are listed in Appendix A. DEGs between osmotic- and salt-treated and control samples were identified using DEseq2 R packages [70] based on the read counts with the adjusted q value < 0.05 and 2-fold change in gene expression. We constructed a weighted gene co-expression network using the WGCNA package in the R program [71]. Using the standardized gene expression matrix as input, a total of 18 samples (including three different treatments: WT and overexpression lines, each with three replicates) were included. GO enrichment analyses were conducted using the ClusterProfiler R package v4.10.0 with DEGs, while GO terms with corrected *p*-values less than 0.05 were considered significantly enriched. Also, the ClusterProfiler R package was used to test the statistical enrichment of DEGs in KEGG pathways. All DEGs were selected for the WGCNA. After threshold screening, a soft threshold power of β = 12 was applied to the original scale-free relationship matrix to obtain an unscaled adjacency matrix. To assess the correlation of gene expression patterns more effectively, the adjacency matrix was further transformed into a Topological Overlap Matrix (TOM), and a dynamic tree-cutting algorithm was employed for gene clustering and module division. The minimum number of genes within a module was set to 30 (minModuleSize = 30), and the threshold for merging similar modules was 0.25 (cutHeight = 0.25). The raw transcriptome reads were submitted to the China National GeneBank database (https://www.cngb.org, accessed on 15 December 2023) under the accession number CNP0004758.

### 4.3. RNA Sequencing (RNA-Seq) Analysis

Total RNA was isolated from 100 mg of control and treated plant samples using a MiniBEST Plant RNA Extraction Kit (TaKaRa, Osaka, Japan); 1 ug RNA was then used for cDNA synthesis with the PrimeScriptTM RT Reagent Kit (TaKaRa, Dalian, China). RT-qPCR was performed using a CFX96 (Bio-Rad, Irvine, CA, USA) with TB Green Premix Ex Taq^TM^ II (Tli RNaseH Plus) (TaKaRa, Dalian, China) according to the manual. Three biological and three technical replicates were used for each sample and AtTubulin (AT1G50010) and AtUbiquitin (AT1G55060) were used as internal reference genes. Relative quantitation was calculated by the 2^−ΔΔCT^ method [29]. All RT-qPCR primers were listed in Appendix A.

### 4.4. Cis-Acting Promoter Element Analysis

According to the *ScDREB10*-overexpression transcriptome data, we selected the most significant differentially expressed up-regulated genes in transgenic plants compare with the WT under different treatments and the analysis methods were performed according to a previously published study from Maruyama et al., 2012 [30]. Then, we downloaded the 1-kb promoters (−1 to −1000) of up-regulated genes from the TAIR database (https://apps.araport.org/thalemine/begin.do, accessed on 23 September 2023). The observed frequencies of all the hexamer (4^6^ = 4096) sequences in the 1-kb promoters of up- regulated genes were counted. Single random sampling (*n* = 100) of 1000 replicates was used to determine standardized promoters. We calculated the sample means of the frequencies of each hexamer sequence in the standardized promoters, and considered those sample means as observed frequencies in the standardized promoters. Scatter plots showed the Z scores (*y*-axes) for the observed frequencies of all hexamer sequences (*x*-axes) in the 1-kb promoters of up-regulated genes in each transgenic plant under different treatments.

### 4.5. Yeast One-Hybrid Assay

For Y1H, the ScDREB10 CDS without a stop codon was amplified and integrated into the sites of the pGADT7-Rec2 vector to form the prey sequence. The specific primers are listed in Appendix A. Triple tandem copies of 13 elements were individually inserted into pHIS2 vectors; the bait sequence is listed in Appendix A. The prey and baits were co-transformed into yeast strain Y187 according to the manual provided by Yeastmaker™ Yeast Transformation System 2 (Clontech, Dalian, China). The co-transformed yeast cells were examined on the SD/-Trp/-Leu and SD/-Trp/-Leu/-His dropout media with the addition of 90 mM 3-amino-2,4-triazole (3-AT). Yeast cells were incubated at 30 °C and observed after 3 days. 

### 4.6. Quantification and Statistical Analysis

For all experiments, the results are shown as means ± SE and statistical significance was determined via one-way ANOVA (LSD) analysis using SPSS (version 21.0, Armonk, NY, USA). Variations were considered significant if * *p* < 0.05 and ** *p* < 0.01. 

## 5. Conclusions

In this study, to delve deeper into the molecular functional mechanism and modulate the physiological pathways governed by ScDREB10, we leveraged the transcriptome data from overexpression-*ScDREB10 Arabidopsis* plants. We performed comprehensive analyses and the results showed that the “phenylpropanoid biosynthesis” and “starch and sucrose metabolism” were key pathways in the network of cyan and yellow modules, and demonstrated the highest enrichment in response to osmotic and salt stress, respectively. The qRT-PCR results confirmed that most genes related to phenylpropanoid biosynthesis” and “starch and sucrose metabolism” pathways in overexpression-*ScDREB10 Arabidopsis* were up-regulated, and the corresponding lignin, sucrose, and trehalose contents and sucrose phosphate synthase activities were also increased in overexpression-*ScDREB10 Arabidopsis* under osmotic and salt stress treatments. Additionally, ScDREB10 was not only able to bind to classical cis-elements, such as DRE and TATCCC (MYBST1), but also bind to unknown elements, like CGTCCA. All the results showed that ScDREB10 may regulate phenylpropanoid biosynthesis, and starch and sucrose metabolism pathways under osmotic and salt stresses, respectively, thus comprehensively improving stress tolerance. This research provides insights into the molecular mechanisms underpinning ScDREB10-mediated stress tolerance and contributes to obtaining a deep understanding of the A-5 DREB regulatory mechanism.

## Figures and Tables

**Figure 1 plants-13-00205-f001:**
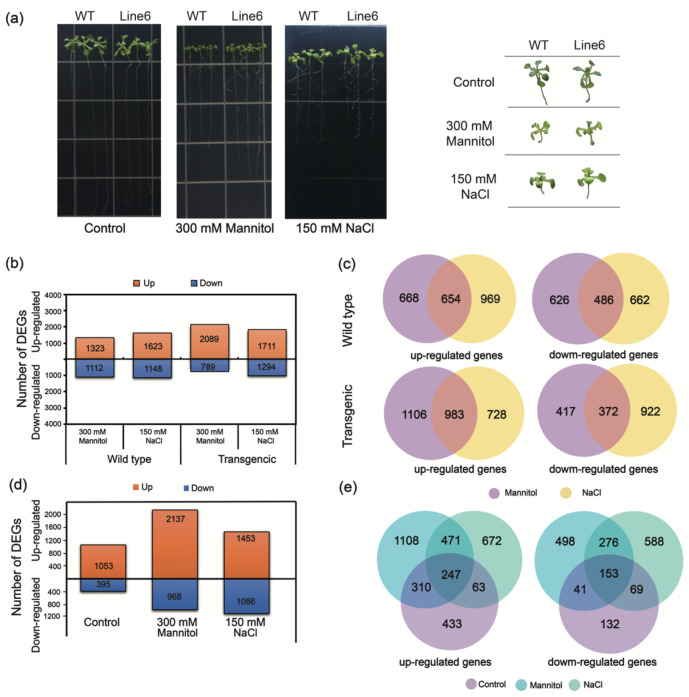
The DEGs analysis between overexpression-*ScDREB10* and WT *Arabidopsis* transcriptomes in response to osmotic and salt stresses. (**a**) The phenotype of overexpression-*ScDREB10* in response to osmotic and salt stresses; (**b**) the total number of up- and down-regulated genes of WT and transgenic lines under osmotic and salt stress treatments compared to control; (**d**) the total number of up- and down-regulated genes of ScDREB10 line compared to WT under control, osmotic and salt stress treatments; (**c**,**e**) Venn diagram showing overlap of up- and down regulated genes analysis between overexpression-*ScDREB10* and WT under control, osmotic or salt stresses corresponding to (**b**,**d**). Genes with adjusted *p* value (FDR) < 0.05, and |log2 fold change| > 1 were listed as significant DEGs.

**Figure 2 plants-13-00205-f002:**
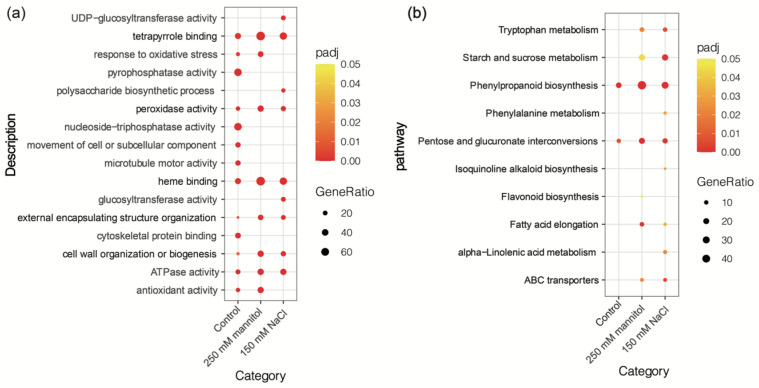
The results of GO enrichment and KEGG pathway analysis for up-regulated DEGs between overexpression-*ScDREB10* and WT transcriptomes in response to osmotic and salt stresses. The figure includes the GO enrichment analysis (**a**) and KEGG pathway analysis (**b**) results. The *X*-axis displays the GeneRatio, while the *Y*-axis represents the terms of pathways. The coloring in the figure is correlated with the padj value, where a lower padj value indicates a more significant enrichment. Additionally, the point size in the figure correlates with the numbers of DEGs.

**Figure 3 plants-13-00205-f003:**
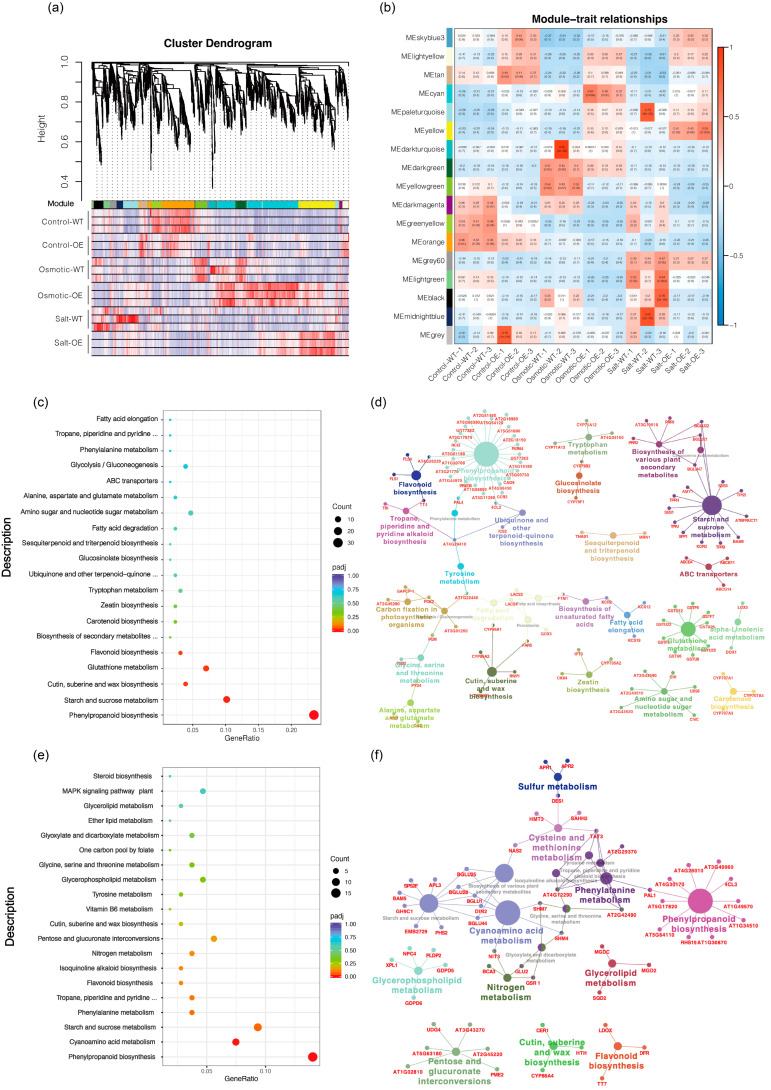
WGCNA analysis to assess all DEGs between the overexpression-*ScDREB10* and WT transcriptomes in response to osmotic and salt stresses. (**a**) A hierarchical cluster tree was generated to identify co-expression modules through WGCNA. Each leaf on the tree represents an individual gene. The tree consists of 17 modules, which are labeled with different colors. (**b**) Heatmap of the relationships between module traits and different lines subjected to various treatments. Each row in the table corresponds to a consensus module, labeled with a color as shown in (**a**), and each column represents a repeat line under different treatments. (**c**) KEGG pathway enrichment analysis was performed for the cyan module. (**d**) Gene network based on the KEGG pathway enrichment analysis for the cyan module. (**e**) KEGG pathway enrichment analysis of yellow module. (**f**) Lastly, a gene network was generated based on the KEGG pathway enrichment analysis for the yellow module. The *x*-axis represents the GeneRatio, while the *y*-axis represents the terms of pathways. The coloring of the plot corresponds to the padj value, where a lower padj value indicates a higher significance of enrichment. The size of the points in the plot correlates with the number of DEGs.

**Figure 4 plants-13-00205-f004:**
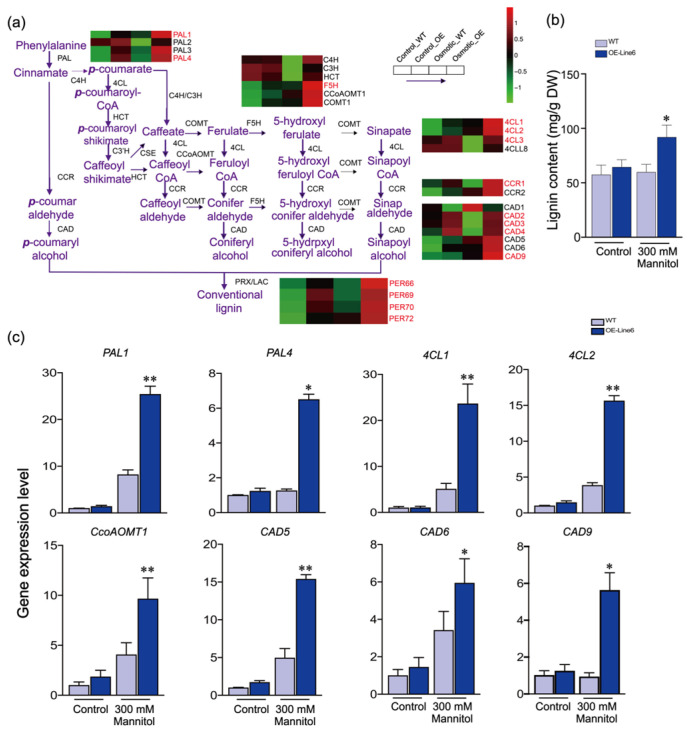
The expression of lignin biosynthesis pathway in overexpression-*ScDREB10* lines and the WT under osmotic stress. (**a**) The pathway and heat map analysis of lignin biosynthesis of overexpression-*ScDREB10* and WT. The heat map analysis revealed that genes in overexpression-*ScDREB10* lines were either upregulated (represented by red bars), not significantly changed (represented by black bars), or downregulated (represented by green bars) under osmotic stress. Red markings indicated significantly increased gene expression levels in overexpression-*ScDREB10* lines compared to WT under osmotic stress. The heat map was constructed using TB Tools [28]. (**b**) qRT-PCR of overexpression-*ScDREB10* and WT under normal conditions and osmotic stress treatment. The WT samples were represented by a light blue box, while the transgenic line 6 samples were represented by a deep blue box. *AtUBQ10* and *AtTubulin* served as reference genes for normalization as described by Choi et al. (2014). The relative gene expression levels were calculated using the 2^−ΔΔCt^ method (Livak and Schmittgen, 2001 [29]) and presented as fold change (FC) compared to the control, which was set at 1. Statistical significance was determined by LSD (* *p* < 0.05, ** *p* < 0.01). (**c**) The lignin content of overexpression-*ScDREB10* and WT under normal conditions and osmotic stress treatment.

**Figure 5 plants-13-00205-f005:**
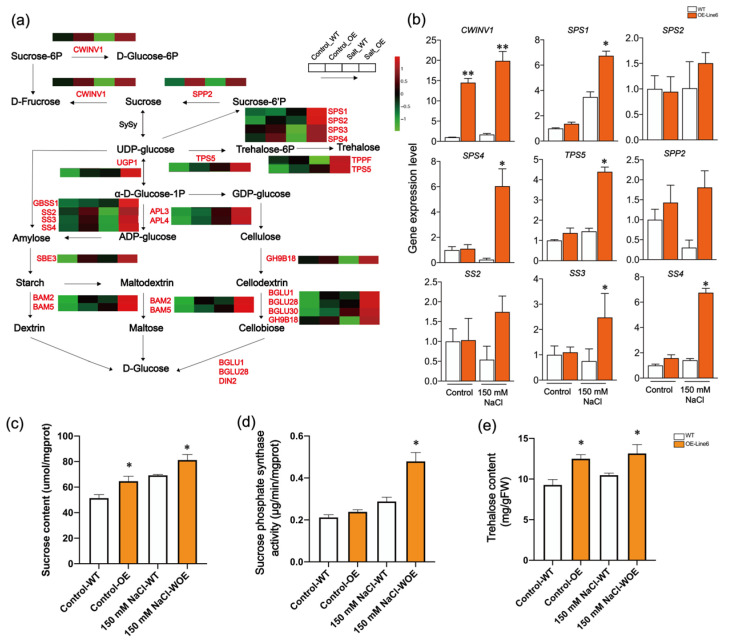
The starch and sucrose metabolism of *ScDREB10*-OE lines and WT under salt stress. (**a**) Heatmap of expression level of genes that are involved in the starch and sucrose metabolism and are distinctly expressed in overexpression-*ScDREB10 Arabidopsis* compared to WT plants. The red and green color indicate up- and down-regulated DEGs, respectively; (**b**) qRT-PCR of overexpression-*ScDREB10* and WT under normal condition and salt stress treatments; the sucrose content; (**c**) sucrose phosphate synthase activity (**d**) and trehalose content (**e**) of overexpression-*ScDREB10* and WT under normal conditions and salt stress treatments; White box indicates the WT and orange box represents the transgenic line 6. *AtUBQ10 (At4g05320)* and *AtTubulin (At1g50010)* were used as reference genes for normalization (Choi et al., 2014). The FC in relative gene expression level was evaluated utilizing the 2^−ΔΔCt^ approach, as described by Livak and Schmittgen (2001) [29], with control values set at 1. The statistical significance (* *p* < 0.05, ** *p* < 0.01) was determined using the LSD test.

**Figure 6 plants-13-00205-f006:**
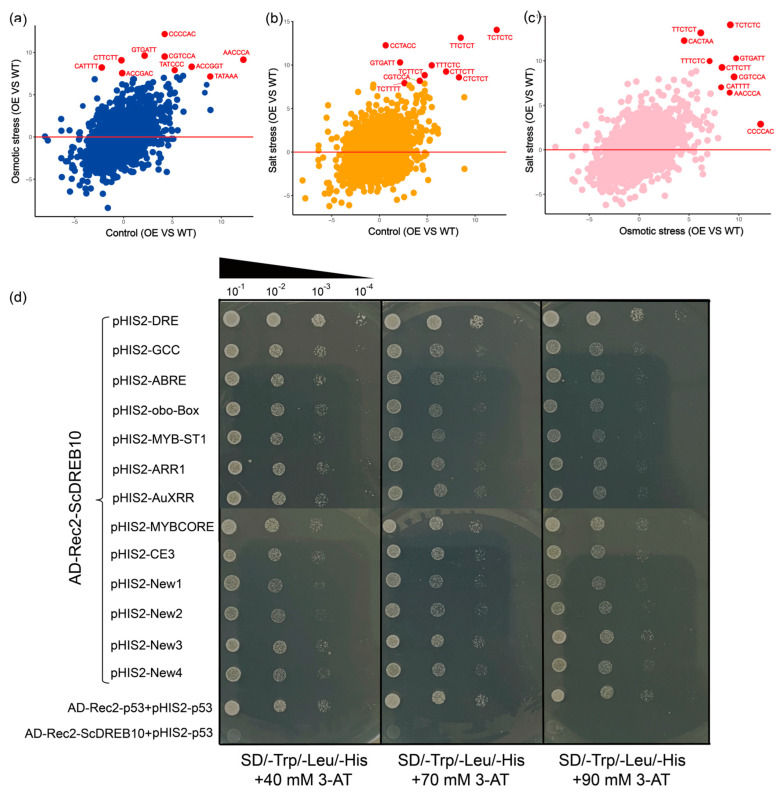
The cis-acting promoter element analysis of ScDREB10 in response to osmotic and salt stresses. (**a**–**c**) The cis-acting promoter element analysis of ScDREB10 in response to osmotic and salt stresses by transcriptome; the hexamer sequence frequencies of up-regulated genes in ScDREB10 were compared to standardized promoters using scatter plots. The number of hexamer sequences was calculated and standardized for every 100 promoters in three comparisons: osmotic stress OE vs. WT vs. control OE vs. WT, salt stress OE vs. WT vs. control OE vs. WT, and osmotic stress OE vs. WT vs. salt stress OE vs. WT. Cis-acting promoter element sequences marked by red indicated the 10 most-frequent hexamers identified among three comparisons; (**d**) determination of the ability of ScDREB10 to bind cis-acting promoter elements by yeast one-hybrid assay. Triple tandem copies of 13 representative cis-element sequences (DRE, GCC, ABRE, obo-Box, MYB-ST1, ARR1, AuXRR, MYBCORE, CE3, New1, New2, New3 and New4) were used as baits. Yeast cells carrying pGADT7-ScDREB10 and 13 pHIS2-cis-acting elements were plated on SD/-Trp/-Leu/-His dropout medium with 40 mM, 70 mM, and 90 mM of 3-AT, respectively. PGADT7-p53 binding to pHIS2-p53 (AD-Rec2-p53 + pHIS2-p53) was used as a positive control, while AD-Rec2-ScDREB10 and pHIS2-p53 were used as negative controls.

## Data Availability

The datasets used and/or analyzed during the current study are available from the corresponding authors upon reasonable request.

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
