# Peer review of "Transcriptome Profiles Reveals ScDREB10 from Syntrichia caninervis Regulated Phenylpropanoid Biosynthesis and Starch/Sucrose Metabolism to Enhance Plant Stress Tolerance"

_plants, 2024, doi:10.3390/plants13020205_

Round 1
Reviewer 1 Report
Comments and Suggestions for Authors
In the present study, the function of ScDREB10 from Syntrichia caninervis is well investigated and several new insights related to its interactions and regulation are provided. In my opinion, this manuscript has high potential for publishing in PLANTs. Other comments:
- Gene name and Arabidopsis as well as scientific names must be italicized. Please consider the entire text.
- All figures are well illustrated.
- Conclusion is not complete, please add more key findings.
Author Response
Dear reviewer:
Thank you for your question. According to your comments and suggestions we have fully considered the comments and suggestions and revised previous version throughout. The point by point responses are followed:
Q1: Gene name and Arabidopsis as well as scientific names must be italicized. Please consider the entire text.
Response: We have carefully checked and revised all gene name and Arabidopsis as well as scientific names throughout the whole manuscript.
Q2: Conclusion is not complete, please add more key findings.
Response: Thank you for your suggestions, I revised the conclusion part, please see the line 523-542 in PDF file.
Reviewer 2 Report
Comments and Suggestions for Authors
Dear Collegues
It was a great pleasure to get acquainted with the submitted manuscript of Yuqing L and et al. The topic is extremely relevant. The aim of the work and the objectives are well formulated and justified. Transgenic Arabidopsis plants overexpressing the transfactor gene ScDREB10 of the moss Syntrichia caninervis have been successfully constructed. This gene is responsible for dehydration tolerance. The choice of this gene is very important because higher plants are unable to tolerate deep dehydration. The resulting transgenic Arabidopsis plants could be expected to have increased tolerance to salt and osmotic stress.
The authors compared the transcriptome profiles of genes from wild-type plants and a transgenic plant containing an overexpressed moss transfactor gene, ScDREB10, under salt and osmotic stressors. Using different methodological approaches, the authors have convincingly demonstrated that the moss transfactor ScDREB10 causes stress-dependent stimulation of the expression of genes involved in phenolpropanoid biosynthesis and starch/sugarose metabolism. In contrast to the vast number of similar articles, the authors did not limit themselves to comparing only the transcriptomic profiles of the analyzed variants. To confirm the RNA-Seq results, RT-qPCR was performed for a number of genes related to starch, sucrose, and phenylpropanoid synthesis. Moreover, it was shown that under the action of salt and osmotic stresses in transgenic plants there was an increase in the content of starch, some sugars and lignin, as well as stimulation of the activity of a number of enzymes involved in carbohydrate metabolism. This suggests that the moss transfactor ScDREB10 regulates not just the expression of phenylpropanoid and sugar biosynthesis genes, but controls these biosynthetic pathways themselves.
I did not find any shortcomings in this paper. I believe that the manuscript can be published as submitted.
Author Response
Dear reviewer,
Thank you for your review. I have made revisions based on the feedback provided by other reviewers. These revisions include improving the clarity and flow of the abstract and conclusion sections. Additionally, I have enhanced the quality of the images and added quality control and mapping rate statistics for the transcriptome data, thereby improving the overall quality of the article. If you have any further inquiries, please do not hesitate to contact me. Thank you once again.
Reviewer 3 Report
Comments and Suggestions for Authors
MS titled "Transcriptome profiles reveals ScDREB10 from Syntrichia caninervis regulated phenylpropanoid biosynthesis and starch/sucrose metabolism to enhance plant stress tolerance" is a well-written, interesting issue. I only have a few minor suggestions for improving the MS considering the size and quality of the figures. Details are in the attached file.
I suggest to accept MS for publication after minor revision.

Author Response
Dear reviewer:
Thank you for your question. According to your comments and suggestions we have fully considered the comments and suggestions and revised previous version throughout. The point by point responses are followed:
Q1: Increase figures, please
Response: Thank you for your suggestions, we improved the figures quality and size in the manuscript.
Reviewer 4 Report
Comments and Suggestions for Authors
Article entitled “Transcriptome profiles reveals ScDREB10 from Syntrichia caninervis regulated phenylpropanoid biosynthesis and starch/sucrose metabolism to enhance plant stress tolerance” is interesting but I have few concerns stated below.
1. Rewrite abstract by highlighting and summarising main findings instead of naming tools and databases used in analysis.
2. In introduction section, why author is highlighting previous study, it should be elaborated in discussion section only. It seems this study is the extension of previous study in which same tools and databases were used as you mentioned in Abstract?
3. Data details for bioinformatics analysis such as assembly and mapping with average length of reads is missing or done in previous study, if so, provide in supplementary table again to avoid data manipulation.
4. Why author used weight gene co-expression network analysis instead of regulatory network to highlight negative or positive regulation of stress response with environment. Explain.
Author Response
Dear reviewer:
Thank you for your question. According to your comments and suggestions we have fully considered the comments and suggestions and revised previous version throughout. The point by point responses are followed:
Q1: Rewrite abstract by highlighting and summarising main findings instead of naming tools and databases used in analysis.
Response: Thank you for your suggestions. I revised it. Please see the abstract section.
Q2: In the introduction section, why author is highlighting previous study, it should be elaborated in discussion section only. It seems this study is the extension of previous study in which same tools and databases were used as you mentioned in Abstract?
Response: Thank you for your question. ScDREB10 was screened from the transcriptome of S. caninervis during dehydration-rehydration, and this current study focuses on delving deeper into possible regulatory mechanisms of ScDREB10 by using the transcriptome of transgenic Arabidopsis. The use of transcriptome data in this article is a novel approach and has not been previously explored. Although these two studies are related, they are not identical. Based on your suggestions, we removed some part of previous study in the discussion section, please see the INTRODUCTION and DISCUSSION section (Line 77-82 and line 313-323 in PDF file).
Q3: Data details for bioinformatics analysis such as assembly and mapping with average length of reads is missing or done in previous study, if so, provide in supplementary table again to avoid data manipulation.
Response: Thank you for your suggestions. As mentioned above, the use of transcriptome data in this article is a novel approach and has not been previously explored. I provided the quality control and mapping with average length of reads file in the supplementary Table S8.
Q4: Why author used weight gene co-expression network analysis instead of regulatory network to highlight negative or positive regulation of stress response with environment. Explain.
Response: Thank you for your question. The choice of using weighted gene co-expression network analysis (WGCNA) over a regulatory network in this study was driven by several key considerations. WGCNA allows for the identification of clusters (modules) of highly correlated genes, enabling us to capture the complex patterns of gene expression under stress conditions more effectively. This approach is particularly advantageous in highlighting both positive and negative correlations in gene expression.
Unlike traditional regulatory network analyses, which often focus on direct interactions and regulatory relationships between specific genes and transcription factors, WGCNA provides a broader, more holistic view of gene interactions. This is particularly important in the context of stress response, where multiple genes and pathways are often simultaneously activated or repressed in response to environmental changes.
By using WGCNA, we can identify not only the individual genes that are central in the response to stress but also the gene modules that show coordinated expression patterns. This can reveal new insights into the complex regulatory mechanisms underlying stress responses, including both author used weight gene co-expression network analysis upregulation and downregulation of genes in response to environmental cues.
So we choose the WGCNA analysis.
Round 2
Reviewer 4 Report
Comments and Suggestions for Authors
This version is better but author should explore that module can be identified in regulatory networks as well as these are not strict for WGCNA.